# Reducing Cement Clinker Sintering Temperature Using 
Fluorine-Containing Semiconductor Waste

**DOI:** 10.3390/ma18174202

**Published:** 2025-09-08

**Authors:** Bilguun Mend, Youngjun Lee, Jang-Ho Jay Kim, Yong-Sik Chu

**Affiliations:** 1Climate and Energy R&D Group, Korea Institute of Ceramic Engineering and Technology, Jinju 52581, Republic of Korea; bilguun@kicet.re.kr (B.M.); uyi047@kicet.re.kr (Y.L.); 2School of Civil and Environmental Engineering, Yonsei University, Yonsei-ro, Seodaemun-gu, Seoul 03722, Republic of Korea

**Keywords:** fluorine mineralizer, semiconductor sludge, cement clinker, energy savings, CO_2_ emissions

## Abstract

This study investigated the potential use of fluorine-containing semiconductor industrial sludge as a mineralizer in Portland cement clinker production. Raw mixes were prepared by partially replacing raw materials with 6%, 9%, and 12% sludge and sintered between 1300 and 1500 °C. The clinker burnability, phase composition, and chemical integrity were evaluated through FreeCaO measurements, X-ray diffraction (XRD) with Rietveld refinement, and X-ray fluorescence (XRF) analyses. The results showed that sludge addition reduced the sintering temperature by up to 150 °C, enabling near-complete clinker formation at 1300 °C for blends containing 9% and 12% sludge (FreeCaO ≤ 0.6 wt.% compared to 62 wt.% in the reference sample). Fluorine incorporation stabilized the re-active β–C_2_S polymorph and shifted the alite (C_3_S) phase distribution from stable M1 to metastable M3 and T3 phases. Additionally, the C_3_A phase content decreased, and a unique fluorine-containing phase, Al_7_Ca_6_O_16_F, formed, promoting clinker formation. Lowering the sintering temperature led to energy savings of 15–22.5% and reduced CO_2_ emissions by 0.10–0.20 tons per ton of clinker, positively impacting the environment. This study demonstrates that recycling industrial sludge can enhance cement production efficiency and support environmental sustainability.

## 1. Introduction

The cement industry, a significant contributor to global CO_2_ emissions, is in urgent need of alternative raw materials and innovative technologies. Accounting for approximately 8% of total anthropogenic releases, the industry’s high temperatures (1450 °C) for clinker production demand enormous amounts of thermal energy [1,2]. The reliance on natural raw materials such as limestone and clay further exacerbates environmental impacts [3], including landscape disruption, biodiversity loss, dust emissions, and groundwater depletion. Increasing energy consumption and the depletion of limited resources underscore this urgent need for exploration. The time to act is now to reduce both the carbon footprint and the energy intensity of cement manufacturing, highlighting the importance of alternative raw materials and sustainable practices [4].

Over the past decades, various industrial by-products have been successfully utilized as partial substitutes for cement production. Fly ash, blast furnace slag, silicomanganese slag, and sewage sludge, for instance, have been incorporated into raw mixes, achieving resource conservation and reducing CO_2_ emissions [5,6]. These practices align with the principles of the circular economy, demonstrating that waste streams can be transformed into valuable resources and integrated into sustainable construction practices [7,8].

Recently, semiconductor industry waste, particularly chemical mechanical polishing (CMP) sludge, has attracted increasing attention. This sludge, despite being classified as hazardous waste, holds significant potential. Its high concentrations of CaO, Al_2_O_3_, and fluorine (F) can significantly influence clinker mineralogy and sintering behavior [9]. The high fluorine content imparts fluxing properties that may lower the melting temperature of clinker raw mixes and promote phase formation [10,11]. This potential of semiconductor-derived fluorine-rich sludge as a mineralizing agent is a beacon of hope in the quest for sustainable cement production [12,13].

Most previous studies have focused on conventional mineralizers (such as CaF_2_ or cryolite) and general industrial by-products [14,15], with limited emphasis on the mechanistic role of fluorine-rich sludge in clinker phase transformations, mineralogical stability, and energy savings [16,17,18]. This research gap underscores the need to clarify not only the chemical contribution of such sludge but also its impact on phase equilibrium, silicate and aluminate competition, and reaction kinetics during clinkerization.

In this context, the present study investigates the incorporation of semiconductor industrial sludge containing 11.4 wt.% fluorine into Portland cement clinker production. Raw mixes were prepared with 6%, 9%, and 12% replacement levels and sintered at temperatures ranging from 1300 °C to 1500 °C. The main objectives were to identify the optimal parameters for significant temperature reduction, evaluate the mineralogical and chemical transformations induced by sludge addition, and quantify the resulting energy savings and CO_2_ emission reductions.

This study not only demonstrates the feasibility of valorizing semiconductor sludge as a novel mineralizer but also provides essential insights for potential industrial-scale applications and long-term performance assessment. The findings, which highlight a promising pathway to reduce the carbon footprint of cement manufacturing, are a significant step forward in the advancement of waste recycling and the development of eco-efficient and sustainable cement production technologies.

## 2. Materials and Methods

### 2.1. Raw Materials

In this study, reagent-grade calcium carbonate (CaCO_3_, Junsei, extra pure), aluminum oxide (Al_2_O_3_), silicon dioxide (SiO_2_), and iron (III) oxide (Fe_2_O_3_) (Samchun, extra pure) were employed as baseline raw materials for clinker preparation. In addition, a fluorine-rich semiconductor sludge, obtained from a local manufacturing facility in South Korea, was incorporated due to its elevated concentrations of calcium oxide (CaO, 36.8 wt.%), aluminum oxide (Al_2_O_3_, 19 wt.%), and fluorine (11.4 wt.%). The sludge was chemically characterized using inductively coupled plasma optical emission spectrometry (ICP-OES), carbon/sulfur analysis, and ion chromatography, with principal oxides expressed as weight percentages in Table 1. CaO and Al_2_O_3_ were expected to enhance the stability of silicate and aluminate phases, while the high fluorine concentration was anticipated to act as a mineralizing flux, lowering melting points and facilitating clinker phase formation. Ordinary Portland Cement (OPC), conforming to KSL 5201 (2013) and exhibiting a compressive strength of 42.5 MPa, was used as the reference material [19,20].

### 2.2. Raw Mix Preparation and Chemical Modulus Calculation

Raw mixes were prepared by partially substituting clinker raw materials with semiconductor sludge at mass replacement levels of 6%, 9%, and 12%. The specific proportions of raw materials and sludge in each blend are provided in Table 2. These substitution levels were determined from preliminary trials and supporting literature: 6% were selected to examine the initial effect of sludge incorporation on clinker properties, whereas 9% and 12% represented incremental levels commonly employed in cement research. Substitutions beyond 12% were excluded to mitigate potential clinker quality deterioration and to avoid practical challenges during the sintering process.

The schematic diagram in Figure 1 illustrates the process of incorporating semiconductor sludge into cement clinker production. In this process, fluorine-rich semiconductor waste is blended with conventional raw materials and then processed through the standard cement manufacturing steps. The presence of semiconductor sludge helps lower the required sintering temperature, which leads to reduced energy consumption and increased efficiency in clinker production. This approach not only offers energy savings but also promotes the utilization of industrial waste, contributing to more sustainable cement manufacturing practices.

To evaluate the suitability of the raw mixes for clinker formation, key chemical moduli were calculated based on oxide weight percentages:Lime Saturation Factor (LSF): Indicates the proportion of calcium oxide relative to silica, alumina, and iron oxide, and is critical for clinker burnability [21].
(1)$$\mathrm{LSF}=\frac{CaO}{2.8\times SiO_{2}+1.2\times Al_{2}O_{3}+0.65\times Fe_{2}O_{3}}\times 100$$Silica Modulus (SM): Ratio of silica to the combined alumina and ferric oxides, impacting the silicate phase formation [22].
(2)$$\mathrm{SM}=\frac{SiO_{2}}{Al_{2}O_{3}+Fe_{2}O_{3}}$$Iron Modulus (IM): Ratio of ferric oxide to alumina, influencing aluminate and ferrite phase content [23].
(3)$$\mathrm{IM}=\frac{Fe_{2}O_{3}}{Al_{2}O_{3}}$$

The calculated moduli for all blended raw mixes were within optimal ranges (LSF ~92.0, SM ~2.5, IM ~1.6), indicating favorable conditions for clinker production.

### 2.3. Sample Preparation and Sintering Procedure

All raw materials were weighed according to the mix proportions specified in Table 2 to ensure sample consistency and reproducibility. The powders were homogenized in an HT-1000 ball mill for one hour using zirconia grinding media of 10 mm and 20 mm in diameter. The use of dual-sized media promoted both impact and attrition mechanisms, thereby enhancing particle size reduction and mixing uniformity.

Following milling, distilled water was gradually added at 35 wt.% of the total powder mass to achieve optimal plasticity for shaping [24]. The resulting paste was pressed manually into spherical pellets of ~25 g each, ensuring consistent size and geometry for uniform sintering behavior. The pellets were then dried at 120 °C for 24 h to eliminate residual moisture and prevent cracking or bloating during high-temperature treatment.

Dried spheres were sintered in a high-temperature electric furnace with an air-heating system, using a heating rate of ~10 °C/min. Sintering was conducted between 1300 °C and 1500 °C in 50 °C increments, after which the specimens were allowed to cool naturally to ~50–60 °C. Each sintering cycle was carefully controlled with respect to heating rate, dwell time, and furnace atmosphere to ensure reproducibility.

For comparison, reference spheres prepared from reference clinkers were subjected to the same sintering conditions, providing a baseline for evaluating the properties and performance of the sludge-incorporated samples.

Figure 2 presents photographs of cement clinker samples following the sintering process. The reference clinker sintered at 1300 °C displays a finer particle morphology, indicative of incomplete sintering. In contrast, the reference clinker processed at 1350 °C exhibits a coarser texture with larger and more aggregated particles, reflecting enhanced sintering progression. Notably, the blended clinker containing 12% semiconductor sludge and sintered at 1500 °C demonstrates a distinctly darker color and the presence of well-defined crystalline structures, suggesting significant phase development and possible alterations in mineralogical composition resulting from the higher temperature and sludge incorporation.

### 2.4. Analytical Methods

#### 2.4.1. Determination of Free Calcium Oxide (FreeCaO)

The content of free calcium oxide, which indicates incomplete clinker formation, was determined using a chemical extraction method. Finely ground clinker was weighed with high precision and mixed with ethylene glycol, a solvent that selectively dissolves this phase. The suspension was incubated in a temperature-controlled water bath at 50–60 °C for about 30 min to ensure complete dissolution while minimizing interference from calcium silicate and carbonate phases. After incubation, the mixture was vacuum filtered to separate the undissolved solids, and the clear filtrate containing the dissolved fraction was carefully collected to avoid loss or contamination [25].

The filtrate was titrated with a standardized 0.1 M hydrochloric acid solution. Bromo-cresol green was used as the pH indicator because it exhibits a distinct color change around pH 4.7–5.2, allowing for the accurate determination of the titration endpoint. During titration, hydrochloric acid neutralizes the dissolved FreeCaO, and the color change from blue to yellow indicates the endpoint [26]. The volume of hydrochloric acid consumed at the endpoint was accurately recorded, and the FreeCaO content was calculated using the stoichiometric relationship between CaO and hydrochloric acid. The calculation accounted for the sample weight and dilution factors to express FreeCaO as a percentage of the clinker sample [27].

#### 2.4.2. X-Ray Fluorescence (XRF) Spectroscopy

The major oxide compositions of clinker samples were quantitatively analyzed by X-Ray Fluorescence (XRF) spectroscopy using a RIGAKU Supermini 200 instrument (RIGAKU, Tokyo, Japan). To prepare the samples for analysis, powdered clinker was pressed into pellets and covered with polyethylene (P.E.) film to prevent contamination and sample loss while maintaining a consistent surface for measurement. This thin P.E. film is X-ray-transparent, allowing for accurate excitation and the detection of characteristic fluorescence without interfering with the analysis [28]. Each sample underwent measurement for 30 min to ensure high precision, accuracy, and reproducibility. The analysis targeted key oxides including calcium oxide (CaO), silicon dioxide (SiO_2_), aluminum oxide (Al_2_O_3_), iron (III) oxide (Fe_2_O_3_), magnesium oxide (MgO), sulfur trioxide (SO_3_), sodium oxide (Na_2_O), and potassium oxide (K_2_O) [29]. The resulting quantitative data were used to verify the chemical composition of the clinker after sintering, confirming that the sintering process achieved the intended oxide proportions [30]. Furthermore, these measurements enabled a direct comparison with the raw mix proportions, providing insight into the consistency and efficiency of the clinker production process.

#### 2.4.3. X-Ray Diffraction (XRD) Analysis

The mineralogical phase identification and quantification of clinker samples were systematically performed using a Bruker D6 Phaser X-ray diffractometer (Bruker, Karlsruhe, German), operated at an accelerating voltage of 40 kV and an emission current of 15 mA. This instrument configuration ensures sufficient X-ray intensity for clear diffraction signals and optimal resolution [31]. To prevent hydration and alteration of the clinker phases during sample preparation, samples were finely ground while suspended in 95% isopropanol [32]. The use of isopropanol as a grinding medium minimizes exposure to atmospheric moisture, thus preserving the original mineral phases for accurate analysis. The powdered samples were then evenly spread on a zero-background sample holder designed to reduce noise and enhance signal clarity [33,34].

Diffraction data were collected by scanning over a 2θ angular range from 5° to 70°, employing a step size typically between 0.02° and 0.05° 2θ, with a total scan time of approximately 3 h. This extended scan duration and acceptable angular resolution enable the acquisition of high-quality diffraction patterns with well-defined peaks, allowing for the precise identification of both major and minor mineralogical phases. For quantitative phase analysis, the Rietveld refinement technique was applied using HighScore Plus software version 3.0.5. The Rietveld method involves fitting a calculated diffraction pattern to the observed data by adjusting crystal structure parameters, peak shapes, background, and instrumental effects [35]. This comprehensive approach facilitates the accurate quantification of phase abundances, even in complex multiphase clinker samples. The resulting phase compositions provide critical insight into clinker quality, informing the optimization of raw material mixtures and sintering conditions to achieve desired cement performance characteristics.

### 2.5. Limitations and Future Work

This study demonstrated the potential of fluorine-rich semiconductor sludge as a mineralizer for reducing clinker sintering temperature; however, several limitations remain. Microstructural characterization techniques such as Scanning Electron Microscopy with Energy-Dispersive Spectroscopy (SEM/EDS), optical microscopy, and Fourier-Transform Infrared Spectroscopy (FTIR) were not employed in the present work. Incorporating optical microscopy in future investigations would allow for a more precise determination of phase size distribution and morphology, thereby complementing the crystallographic data obtained by XRD. Such analyses could provide deeper insights into the mechanisms by which fluorine affects phase formation and stabilization.

Moreover, this research was limited to clinker formation and phase evaluation under laboratory-scale sintering conditions. The hydration behavior, setting time, and mechanical properties of cementitious systems produced with the modified clinkers were not investigated. To establish the practical applicability of this approach, future studies will involve comprehensive mortar- and paste-level performance evaluations, including heat of hydration measurements, compressive strength testing, and durability assessments.

In addition, only natural cooling conditions were applied in this study. Exploring alternative cooling conditions, such as water quenching and controlled slow cooling, in future work will provide valuable insights into how cooling rate influences phase crystallinity, stability, and clinker reactivity.

Finally, the environmental and economic implications of large-scale sludge utilization must be considered. A life-cycle assessment (LCA) and cost–benefit analysis will be essential to validate the sustainability and industrial feasibility of replacing conventional raw materials with semiconductor sludge.

## 3. Results

### 3.1. FreeCaO Analysis

The FreeCaO content of clinker samples incorporating fluorine-containing semiconductor sludge at replacement levels of 6%, 9%, and 12% was systematically evaluated across sintering temperatures ranging from 1300 °C to 1500 °C. The results are summarized in Table 3.

A detailed examination of Table 3 and Figure 3, together with the extended quantitative values presented in Appendix A (Table A1), reveals a pronounced difference in FreeCaO content between the reference OPC and the sludge-blended clinkers, highlighting the impact of fluorine-containing semiconductor sludge on clinker formation kinetics and burnability. At 1300 °C, the reference clinker exhibits a value of 62, indicating highly incomplete mineralization and insufficient solid-state reactions under these conditions. In stark contrast, the blends containing 9 and 12 units of semiconductor sludge show a drastic reduction to 0.4, providing compelling evidence that the fluorine-rich sludge acts as an effective mineralizer. This promotes raw mix reactivity and accelerates both decarbonation and silicate phase formation at lower temperatures.

At 1350 °C, this beneficial effect is further corroborated: the reference clinker retains a relatively high value of 8, whereas the sludge-blended samples achieve near-complete conversion, with values of 0.6 and 0.4 for 9 and 12 units, respectively. The sharp reduction in residual lime in the sludge-containing clinkers across both temperatures suggests enhanced melt formation and greater ionic mobility, facilitating the incorporation of free lime into the principal clinker phases, particularly alite (C_3_S) and belite (C_2_S) [36].

These findings underscore the role of fluorine, introduced via semiconductor sludge, in reducing the activation energy required for key clinker-forming reactions. The fluxing action of fluorine lowers the melting point of the system, expands the temperature window for liquid phase formation, and promotes the earlier onset of mineralogical transformations. Consequently, complete clinker formation can be achieved at temperatures substantially lower than those in conventional practice without sacrificing product quality. Such improvements not only enhance thermal efficiency and reduce fuel consumption but also open new pathways for the sustainable utilization of industrial waste in cement manufacturing.

### 3.2. X-Ray Fluorescence Analysis

The chemical compositions of clinker samples prepared by partially replacing raw materials with fluorine-containing semiconductor sludge at 0%, 6%, 9%, and 12% were analyzed by XRF spectroscopy and compared to the reference clinker. This analysis quantified principal oxides, such as CaO, SiO_2_, Al_2_O_3_, and Fe_2_O_3_, as well as minor oxides including MgO, SO_3_, Na_2_O, and K_2_O, with high precision.

The comprehensive XRF analysis presented in Table 4 reveals systematic variations in clinker chemical composition as a function of semiconductor sludge replacement levels and sintering temperatures. The CaO content, a critical factor influencing clinker phase stability and hydraulic reactivity, remained consistently within the typical Portland cement clinker range (~60–68), demonstrating robust compositional integrity despite partial substitution of raw materials.

Notably, the Al_2_O_3_ concentration increased proportionally with sludge addition, reflecting the high alumina content inherent to the semiconductor sludge. This chemical enrichment is expected to significantly influence clinker phase assemblage, particularly by promoting the formation and volume fraction of calcium aluminate phases (e.g., C_3_A), which can affect cement hydration kinetics and early strength development.

Conversely, the SiO_2_ content remained relatively stable across all samples, thereby maintaining the silica modulus within optimal limits essential for clinker quality. Minor oxides such as MgO, SO_3_, Na_2_O, and K_2_O exhibited minimal fluctuations, indicating that sludge incorporation did not adversely impact these constituents. Notably, the fluorine-rich sludge appears to function as a flux during sintering, lowering the eutectic temperature and facilitating liquid phase formation. This enhances mass transport and accelerates phase transformations, consistent with the reductions in FreeCaO content shown in Figure 3 and the stabilized oxide concentrations illustrated in Figure 4. The fluxing effect of fluorine thus underpins the improved burnability and sintering efficiency observed in sludge-blended clinkers.

The calculated chemical indices, Lime Saturation Factor (LSF), Silica Modulus (SM), and Iron Modulus (IM) remain within optimal ranges for all blends, confirming the chemical viability for producing high-performance clinkers. Collectively, these results demonstrate that the partial replacement of raw materials with semiconductor sludge does not compromise clinker chemical integrity or phase equilibrium but rather introduces beneficial modifications enhancing sintering and clinker quality.

### 3.3. X-Ray Diffraction Analysis

The Clinker samples sintered between 1300 °C and 1500 °C with 6%, 9%, and 12% fluorine-containing semiconductor sludge were quantitatively analyzed by Rietveld refinement of XRD patterns. Figure 5 and Figure 6 show detailed phase distributions, including alite polymorphs (C_3_S M1, M3, T3, R), belite polymorphs (C_2_S α, β, γ), ferrite (C_4_AF), aluminate (C_3_A), free lime, periclase, and a fluorine-containing compound (Al_7_Ca_6_O_16_F).

The reference clinker sintered at 1300 °C showed insufficient alite and belite formation, resulting in a poorly sintered, dusted structure, confirmed by high free lime content in Figure 3 and Figure 5. In contrast, fluorine-rich sludge addition stabilized reactive belite polymorphs and altered alite polymorph distribution, increasing C_3_S M3 and T3 phases.

These changes enable clinker formation at lower temperatures without compromising quality, enhancing sintering efficiency, and reducing cement production energy consumption. Reduced aluminate (C_3_A) content and unique fluorine-bearing phases highlight sludge-derived fluorine’s complex chemical role, potentially improving hydration kinetics and long-term durability.

(a)Fluorine-Induced Modulation of Alite (C_3_S) Polymorph Distribution and Stability

The incorporation of fluorine-rich semiconductor sludge into the raw mix significantly alters the polymorphic distribution of alite phases. Quantitative analysis revealed a marked increase in metastable polymorphs C_3_S M3 and T3 at the expense of the dominant stable C_3_S M1 phase. This transformation suggests that fluorine acts to destabilize the equilibrium favoring C_3_S M1, promoting dynamic phase transitions that enhance crystal nucleation and growth kinetics. Such modulation is critical as polymorphic form influences the hydration reaction rates and, consequently, the early strength development and durability of the cementitious matrix. The dynamic balance among alite polymorphs under fluorine influence likely facilitates more efficient hydration pathways and mechanical performance improvements.

(b)Belite Enhanced Stabilization and Prevalence of Reactive Belite (C_2_S β) Polymorph

The study found a consistent and significant increase in the highly reactive β-polymorph of belite (C_2_S β) with increasing sludge content. This polymorph is known for its superior hydraulic reactivity and contribution to long-term strength gain in cement. Fluorine’s stabilizing effect on this polymorph suggests an alteration of the clinker’s phase equilibrium, enabling enhanced formation and retention of reactive belite at lower sintering temperatures. This stabilization not only supports improved mechanical properties over extended curing periods but also indicates a potential reduction in energy consumption by lowering the required sintering temperature for clinker production [37].

(c)The Impact on Aluminate (C_3_A) and Ferrite (C_4_AF) Phase Chemistry

Fluorine-rich sludge significantly suppressed the formation of the aluminate phase (C_3_A), reducing its content to nearly a quarter of that observed in conventional OPC clinker. Since C_3_A influences setting time and sulfate resistance, this suppression could lead to cement with enhanced durability and tailored setting characteristics. Concurrently, the ferrite phase (C_4_AF) increased in concentration, which affects clinker color and hydraulic properties. The interplay of these phase shifts reflects complex chemical re-balancing induced by fluorine incorporation, which alters the overall clinker mineralogy and its functional properties.

(d)Free Near-Complete Reaction Evidenced by Minimal Free Lime (CaO) Content

The near absence of free lime across all samples, regardless of sludge content, indicates highly efficient sintering and reaction completeness. This observation confirms that fluorine addition does not compromise clinker quality but instead maintains or enhances the complete conversion of raw materials into stable clinker phases. The minimized free lime content is a crucial indicator of clinker stability and long-term performance.

(e)Formation of Unique Fluorine-Bearing Mineral Phase as Mineralizer

The exclusive detection of the A_l7_Ca_6_O_16_F phase in sludge-containing clinkers substantiates the role of fluorine as a mineralizing agent. This unique fluorine-bearing phase integrates into the clinker lattice and promotes phase transformations at reduced sintering temperatures. By lowering the energy barrier for phase formation, this mineralizer facilitates clinker densification and phase homogeneity, ultimately improving clinker microstructure and mechanical integrity.

These fluorine-driven modifications of clinker mineralogy open pathways for energy-efficient cement production by enabling lower temperature sintering without sacrificing product quality. The altered phase assemblage enhances hydration kinetics and mechanical performance, particularly improving early strength gain and long-term durability. This study underscores the potential of using fluorine-containing industrial by-products, such as semiconductor sludge, as functional additives to tailor clinker chemistry, supporting sustainable manufacturing practices and advancing the development of high-performance cementitious materials.

Figure 7 shows powder X-ray diffraction patterns of blended clinkers B6 (black), B9 (red), and B12 (blue) sintered at 1450 °C, compared to a laboratory reference clinker (grey dashed). The major peaks correspond to belite (C_2_S), with minor alite (C_3_S) and a fluoride-stabilized calcium aluminate phase, Ca_6_Al_7_O_16_F. Black arrows indicate Ca_6_Al_7_O_16_F reflections. The inset depicts the crystal structure of Ca_6_Al_7_O_16_F (Ca: blue; AlO_4_ tetrahedra: grey; O/F: red). Intensities are normalized, and patterns are vertically offset for clarity.

### 3.4. Energy Saving and Carbon Emission Reduction Estimation

The incorporation of fluorine-containing semiconductor sludge into clinker production demonstrated the potential to reduce sintering temperature by approximately 100 to 150 °C compared to conventional Ordinary Portland Cement (OPC) clinkers, which typically sinter at around 1450 °C. This temperature reduction directly translates into significant energy savings and associated reductions in carbon dioxide (CO_2_) emissions [38].

(a)Energy Consumption Reduction

Cement clinker production is an energy-intensive process, with fuel consumption strongly dependent on sintering temperature. Previous studies indicate that for every 10 °C decrease in sintering temperature, energy consumption decreases by approximately 1.5%. Applying this relationship to the observed temperature reduction yields an estimated energy saving of the following:(4)$$\Delta E=\frac{100∼150}{10}\times 1.5\%=15\%∼22.5\%$$

(b)Carbon Emission Reduction

Cement manufacturing accounts for approximately 8% of global anthropogenic CO_2_ emissions, primarily due to the combustion of fossil fuels during the production of clinkers. Given that CO_2_ emissions are roughly proportional to energy consumption, the reduction in emissions can be estimated as follows:(5)$$\Delta CO_{2}=\Delta E\times {\mathrm{Current}\;\mathrm{CO}}_{2}\;\mathrm{emissions}\;\mathrm{per}\;\mathrm{ton}\;\mathrm{clinker}$$

As summarized in Table 5, a reduction in clinker sintering temperature by 100 to 150 °C has the potential to decrease energy consumption by approximately 15 to 22.5%. This substantial energy saving directly correlates with a significant reduction in carbon dioxide emissions, estimated to be between 0.105 and 0.203 tons of CO_2_ per ton of clinker produced, assuming emission factors of 0.7 and 0.9 tons of CO_2_ per ton of clinker.

The use of fluorine-rich semiconductor sludge as a mineralizer not only improves clinker quality and promotes phase formation at lower temperatures but also offers a pathway to reduce the environmental footprint of cement production. The potential energy savings and CO_2_ emission reductions contribute to both economic and ecological sustainability, supporting global efforts to mitigate climate change.

## 4. Discussion

This study highlights the significant mineralogical and environmental benefits of incorporating fluorine-containing semiconductor sludge into cement clinker production. Integrating experimental results from XRD, XRF, and FreeCaO analyses provides a comprehensive understanding of fluorine’s impact on clinker phase formation, sintering temperature reduction, and sustainability metrics.

(a)Mineralogical Impacts and Phase Transformations

Rietveld refinement of XRD data showed that fluorine alters the polymorphic distribution of alite (C_3_S), favoring metastable phases C_3_S M3 and T3. This polymorphic shift is likely to positively influence early hydration kinetics and mechanical strength developments [39]. The reactive belite (C_2_S β) polymorph is stabilized and enriched with increasing sludge content, potentially improving long-term strength. Suppression of aluminate (C_3_A) phases in sludge-containing clinkers suggests enhanced sulfate resistance and altered setting behavior [15]. The formation of fluorine-bearing phases such as A_l7_Ca_6_O_16_F confirm fluorine’s role as a mineralizer, facilitating clinker formation at lower temperatures [40]. Although the industrial sludge cannot be classified as a conventional pozzolan due to its limited amorphous silica content, its high CaO and Al_2_O_3_ concentrations actively contributed to clinker phase development, particularly favoring the formation of C_3_S, C_2_S, and C_3_A. The elevated fluorine content further acted as a fluxing agent, lowering the burning temperature of the raw mix by ~150 °C and accelerating the crystallization of clinker minerals. These effects were corroborated by XRF and Rietveld-refined XRD analyses, which confirmed the presence of the expected silicate and aluminate phases.

At clinkerization temperatures, the sludge primarily decomposed into its oxide constituents (CaO and Al_2_O_3_), thereby influencing the chemical balance of the raw mix. While the excess CaO and Al_2_O_3_ could promote the formation of aluminate phases (C_3_A and C_4_AF) at the expense of silicates (C_3_S and C_2_S), the fluxing action of fluorine enhanced the melt phase and accelerated silicate crystallization. Consequently, clinker samples with 6%, 9%, and 12% sludge additions exhibited a stable phase assemblage in which silicate minerals remained dominant, demonstrating that the fluxing role of fluorine effectively mitigates the Ca–Al/Ca–Si competition.

(b)Environmental and Energy Implications

A reduction in clinker sintering temperature by approximately 100–150 °C is estimated to yield energy savings of about 15–22.5%. Considering the inherently high energy demand of clinker production, such a decrease translates into significant operational cost reductions as well as lower fossil fuel consumption [41]. Furthermore, based on reported emission factors of 0.7–0.9 tons of CO_2_ per ton of clinker, this temperature reduction corresponds to a decrease of approximately 0.1–0.2 tons of CO_2_ emissions per ton of clinker produced. This outcome highlights the potential of the proposed approach to mitigate the carbon footprint of the cement industry substantially [17].

Beyond direct reductions in fuel consumption and CO_2_ emissions, lowering the sintering temperature also offers broader environmental benefits. Reduced thermal stress during the production process can minimize NO_x_; emissions, which are another major contributor to air pollution in cement manufacturing [42]. In addition, the partial substitution of conventional raw materials with semiconductor sludge promotes resource efficiency and advances circular economy practices by diverting industrial waste from landfills. This dual advantage reduction of greenhouse gas emissions and waste valorization further enhances the environmental sustainability of clinker production [43].

From an energy perspective, large-scale implementation of this approach could reduce the overall dependency of the cement industry on non-renewable fossil fuels. This is especially relevant under tightening environmental regulations and carbon taxation schemes, where energy efficiency directly translates into financial competitiveness. Future integration of renewable energy sources for kiln operations, combined with sintering temperature reduction strategies, may provide synergistic benefits in advancing the cement sector toward climate neutrality.

(c)Practical Considerations and Future Work

While the benefits of fluorine-containing sludge as a mineralizer are promising, further studies are required to evaluate the long-term durability of cements produced with modified clinker phases. Pilot- and industrial-scale investigations should be conducted to assess kiln operation stability, emission control, and product consistency, ensuring that the approach is technically and economically feasible under real production conditions.

Additionally, detailed research on phase transformation kinetics, hydration mechanisms, and the effects of different cooling regimes will provide deeper insight into the role of fluorine in clinker reactivity. Future work should also extend to evaluating hydration performance, mechanical strength, and durability, supported by life-cycle and cost–benefit analyses to confirm the environmental and industrial sustainability of this method.

(d)Sustainability Considerations of Fluorine Evaporation

During the thermal treatment of the sludge, some fluorine species may volatilize; however, under the experimental conditions of this study, the risk of direct HF emission is considered relatively low. This is because most of the fluorine tended to react with Ca and Al, forming stable crystalline phases such as CaF_2_ or Ca–Al–F compounds, as confirmed by XRF and Rietveld-refined XRD analyses. Therefore, instead of being fully released into the atmosphere, the majority of fluorine remained incorporated into the clinker structure in a stable form.

In addition, fluorine acted as a fluxing agent, lowering the melting temperature of the raw mix and promoting the formation of clinker phases. As a result, the clinker burning temperature was reduced by approximately 150 °C, leading to decreased energy consumption and a significant reduction in CO_2_ emissions. This represents an important contribution to the sustainability of cement production.

Hence, the overall sustainability of sludge recycling must be considered in terms of two aspects:(a)Positive effect reduction of energy demand and CO_2_ emissions(b)Limited risk only a minor fraction of fluorine may volatilize during processing.

Our findings suggest that most of the fluorine remained bound within the clinker in the form of stable phases, and thus, the probability of HF release is relatively low. Nevertheless, future studies should include direct monitoring of possible gaseous emissions and evaluate emission control strategies (e.g., filtration or neutralization systems). In this way, the use of high-fluorine industrial sludge can combine the benefits of improved energy efficiency and CO_2_ reduction while minimizing environmental risks.

## 5. Conclusions

This mineralogical transformation and performance benefits

Fluorine-containing semiconductor sludge, when used as a partial substitute in clinker production, stabilizes the reactive β-polymorph of belite, alters the polymorphic distribution of alite, and promotes the formation of distinct fluorine-bearing phases. These mineralogical changes accelerate early hydration kinetics and improve the mechanical strength development of cementitious materials, confirming the performance advantages of sludge incorporation.

2.Energy and environmental implications

The incorporation of fluorine enables a reduction in clinker sintering temperature of up to 150 °C without compromising phase quality or stability. This thermal decrease results in estimated energy savings of 15–22.5% and a reduction of approximately 0.1–0.2 tons of CO_2_ emissions per ton of clinker, thereby providing clear environmental and operational benefits.

3.Pathway for sustainable cement production

Collectively, the findings demonstrate a promising approach to sustainable cement manufacturing through the valorization of industrial semiconductor waste. Future work should prioritize scale-up studies, long-term durability assessments, and mechanistic investigations of fluorine’s influence on hydration and phase transformation to optimize clinker formulations and advance eco-efficient cement technologies.

## Figures and Tables

**Figure 1 materials-18-04202-f001:**
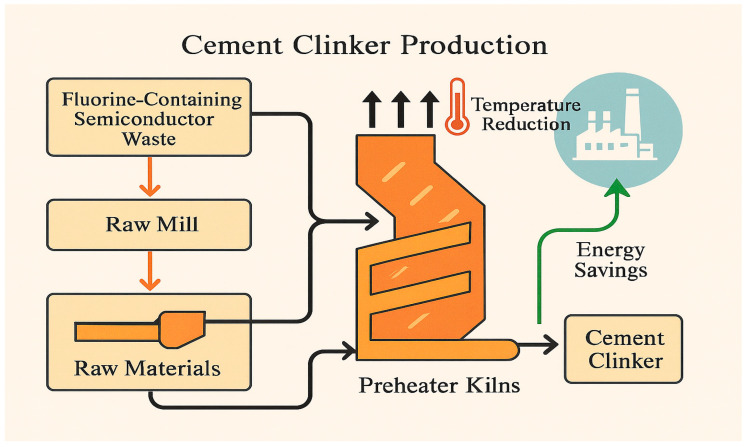
Clinker production process using semiconductor sludge.

**Figure 2 materials-18-04202-f002:**
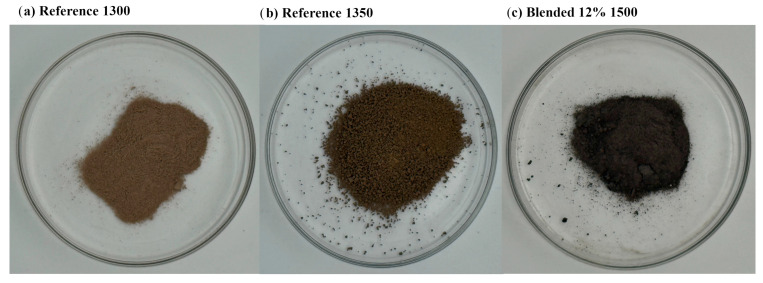
Photograph of dusted clinkers after sintering.

**Figure 3 materials-18-04202-f003:**
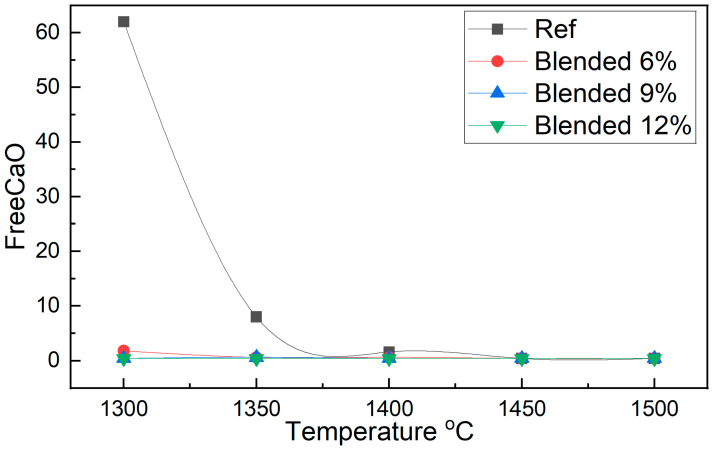
Burnability curves of reference and sludge-blended clinker samples showing FreeCaO content as a function of sintering temperature.

**Figure 4 materials-18-04202-f004:**
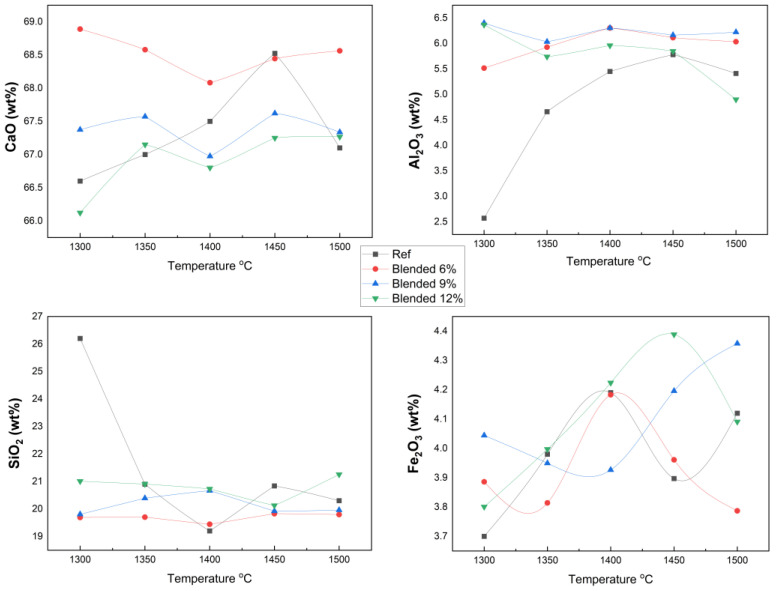
XRF results of principal oxides in clinker samples.

**Figure 5 materials-18-04202-f005:**
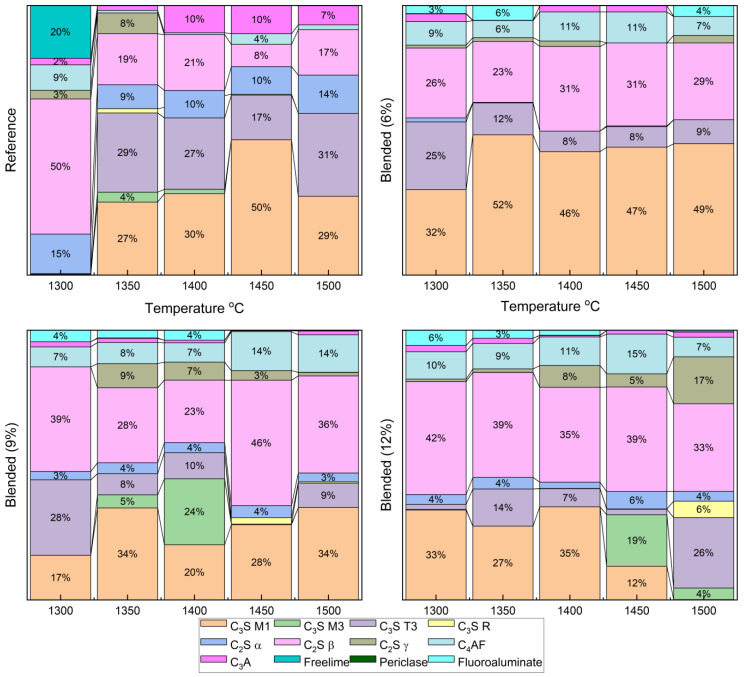
Rietveld refinement of XRD patterns for all clinker samples.

**Figure 6 materials-18-04202-f006:**
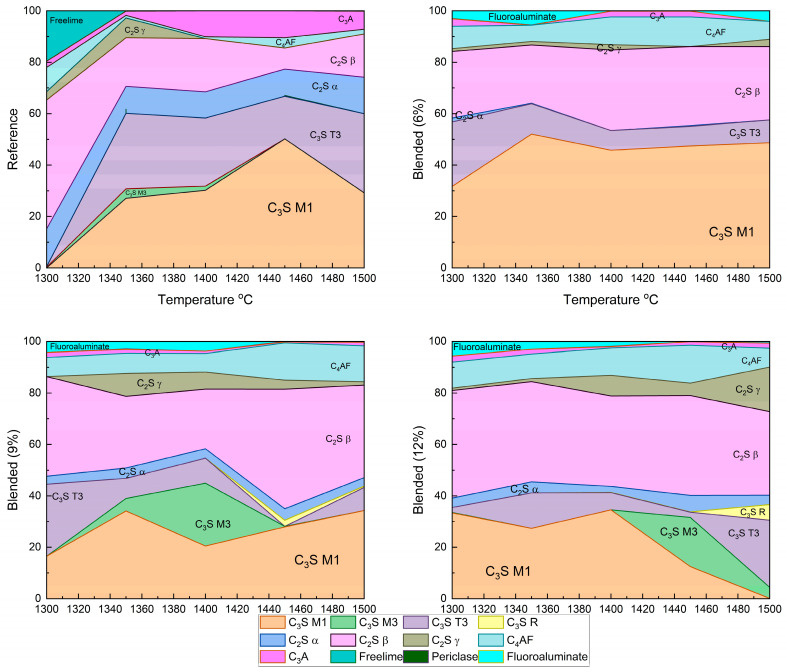
Rietveld refinement of XRD patterns showing phase composition of clinker samples with different sludge contents and sintering temperatures.

**Figure 7 materials-18-04202-f007:**
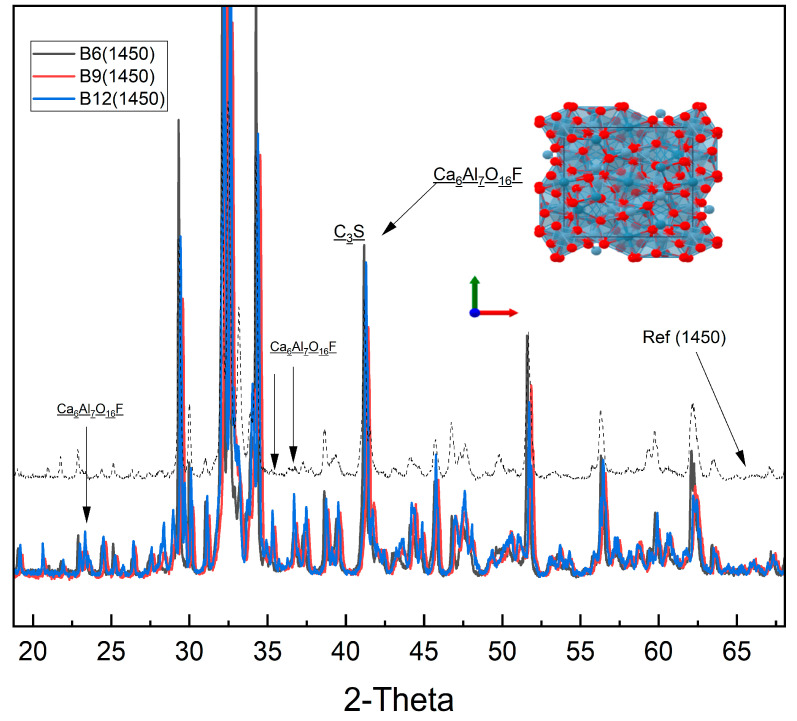
XRD patterns of fluoride-stabilized clinkers sintered at 1450 °C compared to reference clinker.

**Table 1 materials-18-04202-t001:** Chemical composition of semiconductor waste sludge (wt.%).

Sample	SiO_2_	CaO	P_2_O_5_	MgO	F	Fe_2_O_3_	Al_2_O_3_	C	SO_3_	Others	LOI
Sludge (Sc)	3.84	36.8	3.29	3.16	11.4	0.36	19	4.39	3.88	>0.3	28.6

Note: LOI (Loss on Ignition) accounts for volatiles lost during heating; “Others” include trace elements below 0.3 wt.%.

**Table 2 materials-18-04202-t002:** Raw mix design (g).

Sample	CaCO_3_	Al_2_O_3_	SiO_2_	Fe_2_O_3_	Sludge (Sc) %	(F) Content %
Reference	78.88	3.72	15.07	2.33	0	0
Blended 6	74.42	2.55	14.74	2.29	6	0.68
Blended 9	72.19	1.97	14.58	2.27	9	1.03
Blended 12	69.95	1.38	14.41	2.25	12	1.37

**Table 3 materials-18-04202-t003:** FreeCaO results.

Sample	1300 °C	1350 °C	1400 °C	1450 °C	1500 °C
Reference	62	8	1.6	0.4	0.4
Blended 6%	1.8	0.6	0.6	0.4	0.6
Blended 9%	0.4	0.6	0.4	0.4	0.4
Blended 12%	0.4	0.4	0.4	0.4	0.4

**Table 4 materials-18-04202-t004:** XRF analysis results (wt.%).

Samples	CaO	SiO_2_	Al_2_O_3_	Fe_2_O_3_	MgO	P_2_O_5_	SO_3_	Cl	K_2_O	SrO
Ref (1300)	66.60	26.20	2.57	3.7		0.8	0.01		0.06	0.03
Ref (1350)	67.00	20.90	4.66	3.98		0.98	0.01	0.01	0.57	0.04
Ref (1400)	67.50	19.20	5.45	4.19		0.95		0.01	0.06	0.04
Ref (1450)	68.52	20.83	5.77	3.89		0.84		0.01	0.06	0.04
Ref (1500)	67.10	20.30	5.41	4.12		0.88			0.06	0.04
B6 (1300)	68.89	19.69	5.51	3.88	0.59	1.16	0.15	0.01	0.05	0.04
B6 (1350)	68.57	19.70	5.92	3.81	0.51	1.99	0.15	0.01	0.05	0.04
B6 (1400)	68.08	19.44	6.30	4.18	0.65	1.12	0.08	0.01	0.07	0.04
B6 (1450)	68.44	19.82	6.11	3.96	0.34	1.09	0.10	0.01	0.05	0.04
B6 (1500)	68.56	19.79	6.03	3.78	0.41	1.16	0.13	0.01	0.05	0.04
B9 (1300)	67.37	19.80	6.39	4.04	0.63	1.30	0.32	0.01	0.07	0.04
B9 (1350)	67.57	20.39	6.03	3.94	0.32	1.35	0.26		0.07	0.04
B9 (1400)	66.97	20.66	6.30	3.92	0.45	1.22	0.30		0.05	0.04
B9 (1450)	67.62	19.92	6.16	4.19	0.43	1.23	0.27		0.06	0.04
B9 (1500)	67.33	19.96	6.22	4.35	0.51	1.23	0.27		0.06	0.04
B12 (1300)	66.12	21.00	6.36	3.80	0.72	1.36	0.42		0.06	0.03
B12 (1350)	67.14	20.90	5.73	3.99	0.33	1.37	0.39		0.07	0.04
B12 (1400)	66.80	20.72	5.95	4.22	0.39	1.39	0.40		0.06	0.04
B12 (1450)	67.24	20.12	5.84	4.38	0.52	1.36	0.38		0.07	0.04
B12 (1500)	67.26	21.25	4.89	4.09	0.41	1.46	0.46		0.06	0.04

**Table 5 materials-18-04202-t005:** Energy savings and CO_2_ reductions resulting from a decrease in clinker sintering temperature.

Temperature Reduction (°C)	Energy Savings (%)	CO_2_ Emission Reduction (0.7 t CO_2_/t Clinker)	CO_2_ Emission Reduction (0.9 t CO_2_/t Clinker)
100	15	0.105 t	0.135 t
150	22.5	0.158 t	0.203 t

## Data Availability

The original contributions presented in this study are included in the article. Further inquiries can be directed to the corresponding author.

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
