# Peer review of "Reducing Cement Clinker Sintering Temperature Using Fluorine-Containing Semiconductor Waste"

_materials, 2025, doi:10.3390/ma18174202_

Round 1
Reviewer 1 Report
Comments and Suggestions for Authors
In the present manuscript, high fluorine-containing semiconductor industrial waste is considered for reducing cement clinker. After thermal treatments, the waste characteristics are investigated by X-ray diffraction with Rietveld refinement and X-ray fluorescence. The conclusion is that recycling industrial sludge can enhance cement production. After revision, the manuscript can be accepted for publication.
1) The XRF data showed that the main components of the waste are: CaO, Al2O3, and F. How is the pozzolanic characteristic of the material? And how is the impact on the reactivity of the waste due to the high concentration of F?
2) After thermal treatment, the fluor specie is evaporated; more discussion on the sustainability impact should be provided.
3) Incorporations of 6%, 9%, and 12% of waste are considered in the clink preparation; however, in high temperatures, only the oxide compounds are present. The chemical perspective should be discussed. Technically, the addition of the waste after thermal treatment is mainly the addition of CaO+Al2O3 systems, that can eventually compete with the calcium silicate systems.
4) Please review the quantity of decimal numbers in the Table 4, I believe that 2 are the maximum.
5) The Discussion section is short, the authors can merge it with the Results section.
6) More characterizations could be considered, for example, SEM, SEM/EDS, and FTIR to characterize the phases better.
7) The authors employed the sustainable clinkers in cement-based materials, such as mortars? This is important to evaluate the quality of the materials.
Author Response
Dear Reviewer,
We sincerely thank you for taking the time to review and improve our research article. Your valuable comments and suggestions have been carefully considered, and we have attached our detailed responses for your kind review.
Thank you once again, and we wish you a pleasant day.
Sincerely,
Bilguun Mend

Reviewer 2 Report
Comments and Suggestions for Authors
The present research deals with assessing the potential of reducing cement clinker sintering temp using semiconductor wastes. The study is worthy of investigating, not only aligning with the current trend of decarbonising construction materials, but also fitting the scope of this journal. The research is presented with acceptable clarity and replicability, with only 2 minor concerns to be further clarified.
- Is there any alternative choice of semiconductor waste, other fluorine-based, can be used in a similar fashion? The temperature reduction is not significant, which may also be dependent on the characteristics of the waste sources. Thus, it is of interest to clarify this.
- Please conduct thorough proofreading, to make sure the manuscript is free of gramatical error before publication.
Author Response
Dear Reviewer,
We sincerely thank you for taking the time to review our manuscript and provide constructive comments. Your valuable feedback has greatly helped us in improving the quality and clarity of our work.
Please find attached our detailed responses to your comments, along with the corresponding revisions made in the manuscript for your kind review.
We truly appreciate your effort and consideration, and we wish you a pleasant day.
Sincerely,
Bilguun Mend

Reviewer 3 Report
Comments and Suggestions for Authors
The manuscript presents an interesting study on the development and characterization of cement-based composites incorporating waste materials, showing potential for sustainable construction applications.
In the Introduction, the authors provide a relevant background on waste valorization and cementitious materials, but the section could be more concise, avoiding redundancy and integrating more recent references to strengthen the state-of-the-art context.
The research gap is mentioned, yet it would be clearer if directly contrasted with specific limitations of previous studies.
In the Materials and Methods section, the description of sample preparation and testing procedures is detailed, but some parameters (e.g., curing conditions, exact particle size distributions) should be more explicitly defined to enhance reproducibility.
The Results and Discussion are well organized, but certain findings are presented descriptively without sufficient mechanistic explanation; for example, microstructural changes observed via microscopy could be better linked to the mechanical performance trends.
Comparative analysis with existing literature would further validate the novelty of the reported results. Figures and tables are generally clear, although some could benefit from normalization of data for easier cross-comparison, and statistical analysis should be included where applicable.
The Conclusions summarize the main outcomes but would be more impactful if they explicitly addressed the practical implications, scalability, and potential limitations of the developed composites.
Author Response

(The authors gave the same response as above.)

Reviewer 4 Report
Comments and Suggestions for Authors
Reviewer's report on the paper titled “Assessment of the Potential for Reducing Cement Clinker Sintering Temperature Using High Fluorine-Containing Semiconductor Industrial Waste” for the MDPI Journal Materials
The paper “Assessment of the Potential for Reducing Cement Clinker Sintering Temperature Using High Fluorine-Containing Semiconductor Industrial Waste” is an original contribution, which corresponds to the scope of MDPI Journal Materials.
The potential use of fluorine-containing semiconductor industrial sludge as a mineralizer in Portland cement clinker production was investigated in the current study.
The following improvements/complications should be done to made the paper more clear for its understanding:
- The title of the paper seems to be too long and should be shortened.
- The raw mixes content used in the current investigation, should be explained and based in the more details.
- The different substitution levels for investigating the clinker sintering behavior should be explained in the text of the Chapter 1 “Introduction” in the more details. The diapazon of the temperature changes should be based in the more details also.
- The aim formulation should be completed by the namely mentioning of the "optimal parameters for significant temperature reduction".
- The designations in all the figures should be done by the same dimensions of the letters.
- Paragraphs in the sub-chapters 3.3 “X-Ray Diffraction analysis” and 3.4 ” Energy Saving and Carbon Emission Reduction Estimation” should not be numbered by the same manner as the chapters of the paper 1., 2., 3. ...
Author Response

(The authors gave the same response as above.)

Round 2
Reviewer 3 Report
Comments and Suggestions for Authors
In lines 33-35: Mention the environmental impacts of using raw materials.
In lines 38-41: I suggest mentioning that silicomanganese slag is also used, as indicated in the literature: https://doi.org/10.1016/j.conbuildmat.2022.129938
Section 2.1: The authors abbreviate OPC, CaCO3, Al2O3, SiO2, and Fe2O3. I do not see the need to abbreviate them again in section 2.2. Please review the entire manuscript to remove this error, as it also appears in sections 2.3, 2.4.2, etc.
There is a typo in Table 2, where (g) is too far to the right. The same error appears in Table 4.
Author Response
We sincerely thank the reviewer for the constructive comments and helpful suggestions, which have greatly improved the quality of our manuscript. In response to the first comment (Lines 33–35), we have revised the introduction to emphasize the environmental impacts of conventional raw materials, including COâ‚‚ emissions from limestone decarbonation as well as the ecological burden of quarrying and transporting natural resources. Regarding the second comment (Lines 38–41), we have incorporated additional information on silicomanganese slag as an alternative industrial by-product, with reference to the suggested literature (https://doi.org/10.1016/j.conbuildmat.2022.129938), to broaden the context of raw material utilization. For the third point, we carefully reviewed the entire manuscript and removed unnecessary repetitions of abbreviations such as OPC, CaCO₃, Alâ‚‚O₃, SiOâ‚‚, and Feâ‚‚O₃. These corrections were applied not only in Section 2.1 but also consistently in Sections 2.2, 2.3, and the Results section to ensure clarity and uniformity throughout the manuscript. Finally, in accordance with the fourth comment, we corrected the typographical error in Tables 2 and 4 by realigning the unit “(g)” to improve formatting and readability. We believe these revisions adequately address the reviewer’s concerns.